

# The Morphology of Poleward Moving Auroral Forms

**Anton Goertz**[1,2,3], **Noora Partamies**[2,3], **Daniel Whiter**[4], **and Lisa Baddeley**[2,3]

[1]Intelligence and Space Research, Los Alamos National Laboratory, Los Alamos, NM, USA

[2]Department of Arctic Geophysics, University Centre of Svalbard, Longyearbyen, Norway

[3]Birkeland Centre for Space Science, University of Bergen, Norway

[4]School of Physics and Astronomy, University of Southampton, UK

**Correspondence:** Anton Goertz (agoertz@lanl.gov)

**Abstract.** We investigated the morphology of poleward moving auroral forms (PMAFs) qualitatively by visual inspection of all sky camera (ASC) images and quantitatively using the arciness index. The PMAFs in this study were initially identified with a meridian scanning photometer located at Kjell Henriksen Observatory (KHO), Svalbard, and analyzed using ASC images taken by cameras at KHO and in Ny-Ålesund, Svalbard. We present a detailed six-step evolution of PMAF morphology in two dimensions. This evolution includes (1) an equatorward expansion of the auroral oval and an intensification of auroral brightness at the open-closed boundary, (2) the appearance of an arc-like structure in the oval, (3) poleward and possible west/eastward propagation, (4) azimuthal expansion events, (5) rebrightening of the PMAF and eventual (6) fading away. While there have been previous studies commenting on PMAF morphology this is the first work dedicated to the morphological evolution of PMAFs and it includes more detailed discussion and novel aspects, such as the observation of initial merging of $557.7\,\mathrm{nm}$ auroral patches to form a PMAF. Moreover, the morphology of PMAFs is quantified using the arciness index, which is a number describing how arc-like auroral forms appear in ASC images. We present the results of a superposed epoch analysis of arciness in relation to PMAF occurrence. This analysis uncovered that arciness increases suddenly during the onset of a PMAF event and decreases over the PMAF lifetime to return to its baseline value once the event has concluded. This behavior may be understood based on changes in the morphology of PMAFs and the auroral oval and furthermore may be used to identify PMAFs from arciness data.

## 1 Introduction

This study investigates the morphology of specific types of transient dayside auroral features called poleward moving auroral forms (PMAFs). PMAFs are auroral arc-like structures that are aligned along magnetic east-west. They typically span $500 - 1000\,\mathrm{km}$ longitudinally and $\sim 50\,\mathrm{km}$ latitudinally. PMAFs are characterized by poleward motion following their formation inside the auroral oval. The beginning of a PMAF event typically coincides with a brightening of the equatorward or poleward boundary of the auroral oval (equatorward/poleward boundary intensification, EBI/PBI) as well as a slight equatorward expansion of the oval of approximately 0.5° MLAT (Horwitz and Akasofu, 1977; Sandholt et al., 1986; Fasel et al., 1992; Pudovkin et al., 1992; Sandholt and Farrugia, 2007; Frey et al., 2019). Typical poleward propagation speeds lie at around 500 m/s (Oksavik et al., 2005). PMAFs occur exclusively in the dayside auroral oval around magnetic noon (Lockwood et al., 1989), which is a consequence of their formation mechanism. It is widely accepted, that PMAFs are the visual signatures of so called flux transfer events (FTEs) (Sandholt et al., 1986; Fasel et al., 1992; Xing et al., 2012; Lockwood et al., 1989; Wang et al., 2016). Yet, there have also been reports of PMAF events being triggered by short term solar wind dynamic pressure enhancements (Kozlovsky et al., 2005; Maynard et al., 2006).



An FTE occurs during dayside magnetopause reconnection typically following the southward turning of the interplanetary magnetic field (IMF) (Goertz et al., 1985; Neudegg et al., 1999; Wang et al., 2016). Due to the energy enhancement that solar wind particles experience during an FTE, auroral emissions in the $557.7\,\mathrm{nm}$ line are more intense compared to other dayside aurora. PMAFs can nevertheless be observed in both the $630.0\,\mathrm{nm}$ and $557.7\,\mathrm{nm}$ auroral emission lines, among others.

5 PMAFs have an average lifetime of approximately five minutes (Fasel, 1995), and they frequently occur in sequence. In the case of sequential PMAF events, the average time between two consecutive PMAFs is about seven minutes (Fasel, 1995).

As PMAFs are the auroral manifestation of pulsed dayside magnetopause reconnection, they are expected to occur during southward IMF. While PMAF events do seem to occur more often during southward IMF (approximately 60% of 10 events), they also occur during northward IMF (Xing et al., 2012, 2013; Wang et al., 2016). Additionally, the IMF $B_y$ component seems to play a role for the occurrence of PMAFs, as the majority of PMAFs occur during eastward IMF (Xing et al., 2012, 2013), suggesting lobe reconnection might be of relevance.

Particle precipitation leading to auroral emissions associated with PMAFs is due to electrons at hundreds of eV to 1 keV. While that energy range is low compared to nightside aurora, PMAFs are nevertheless the highest energy dayside auroral form 15 (Oksavik et al., 2005; Lorentzen et al., 2010).

In this paper, the morphology of PMAFs is investigated qualitatively and quantitatively. This topic has been addressed in the literature before. Vorobjev et al. (1975) and Horwitz and Akasofu (1977) first documented optical observations of PMAFs and commented on their morphological evolution. They noted that each event coincided with an equatorward 20 expansion of the auroral oval and suggested both phenomena, PMAF occurrence and oval expansions, are causally related. Sandholt et al. (1986) first discovered a dependence of the motion and size of PMAFs on IMF $B_y$. Eastward (westward) IMF would indicate that PMAFs propagate poleward and dawnward (duskward). Furthermore, Sandholt noted that IMF $B_y$ would also determine the direction in which brightness enhancements in PMAFs would spread through the arc, which lined up with the azimuthal motion of the arc. Soon after, it was reported that the equatorward boundary of the auroral oval would intensify 25 during the onset of a PMAF event (Smith and Lockwood, 1990; Lockwood, 1991).

The first complete description of the morphological evolution of PMAF events was published by (Fasel et al., 1992). After analyzing unfiltered all sky television camera data, Fasel divided the evolution of PMAFs into five steps, (1) an equatorward shift of the auroral oval, (2) sudden brightening of the auroral oval, (3) the emergence of an auroral arc out of the oval propagating poleward, (4) localized brightenings spreading along the arc latitudinally, and finally, (5) brightening of the 30 auroral oval at the same time as (4).

Moreover, Fasel et al. (1994) divided PMAFs into three classes based on their exact morphological evolution. PMAF1 events would propagate polewards and slowly fade. PMAF2 events would re-brighten during their poleward motion, while PMAF3 events would re-brighten and slow down. It was found that the large majority of events would fall into the PMAF2 category (84%), followed by PMAF1 events (15%). Only about 1% of PMAFs are PMAF3 events (Fasel, 1995).

35 More recently, Sandholt and Farrugia (2007) investigated the occurrence rate of different classes of PMAFs in the pre-noon and post-noon sectors with regard to steady and positive IMF $B_y$ conditions. It was found, that during eastward IMF pre-noon PMAFs would fall exclusively into the PMAF2 category, while post-noon events would be considered PMAF1s. Additionally, the auroral emission strengths in the $557.7\,\mathrm{nm}$ and $630.0\,\mathrm{nm}$ emission lines at high latitudes were observed to be different in the pre-noon and post-noon sectors. Pre-noon PMAFs (PMAF2s) were observed to have strong $557.7\,\mathrm{nm}$ emissions throughout 40 their entire event lifetime, while post-noon events (PMAF1s) appeared to be fading more quickly in $557.7\,\mathrm{nm}$ emission.

Another pre-/post-noon asymmetry that was discovered by Sandholt relates to auroral oval intensifications at the onset of PMAF events. Under eastward IMF pre-noon events would lead to equatorward boundary intensifications, while post-noon events would not. Instead, Sandholt observed poleward boundary intensifications during the onsets of post-noon PMAF events under duskward IMF. Their findings also confirmed the dependence of the magnetic local time of PMAFs on the IMF $B_y$ 45 direction.

This paper expands upon the previous work and presents a more detailed description of the morphological evolution of PMAFs. We used Meridian Scanning Photometer (MSP) data from the Kjell Henriksen Observatory (KHO) (78.15° N, 16.04° E) in Longyearbyen, Svalbard, to initially identify PMAF events. We made use of All Sky Camera (ASC) images taken in Longyearbyen and Ny-Ålesund (78.92° N, 11.91° E) to analyze the structure and evolution of PMAFs. Furthermore, 50 we quantified the morphology of PMAFs using an index first introduced by Partamies et al. (2014) known as the arciness index. The motivation behind the use of the arciness index was to (1) have a quantitative and objective measure of auroral morphology, and (2) to enable the development of an automated algorithm capable of identifying PMAFs based on arciness data.





## 2 Instrumentation

MSPs are commonly used to study auroral occurrence and movement along the magnetic meridian. The MSP used for this analysis is located at the Kjell Henriksen Observatory (KHO) on Breinosa, Svalbard. The instrument consists of a mirror that scans the meridian along the magnetic north-south direction in a period of 16 seconds. It possesses four channels, that each consist of a narrow bandpass filter mounted onto a tilting frame in front of a photo-multiplier tube. Three of the four bandpass filters are designed to observe aurora emitted by atomic oxygen ($557.7\,\mathrm{nm}$, $630.0\,\mathrm{nm}$ and $844.6\,\mathrm{nm}$). The fourth detector measures photon intensities in a molecular nitrogen emission line ($427.8\,\mathrm{nm}$). The green ($557.7\,\mathrm{nm}$) and red ($630.0\,\mathrm{nm}$) channels are most relevant for this analysis, as auroral emissions associated with PMAFs are strongest in those channels. Both emission lines are caused by forbidden transitions in atomic oxygen with a lifetime of $\tau_{557.7\,\mathrm{nm}} = 0.7\,\mathrm{s}$ and $\tau_{630.0\,\mathrm{nm}} = 107\,\mathrm{s}$, respectively. In this work, the MSP was used to identify PMAF events before each event was analyzed in detail using ASC images.

ASCs capture the entire sky in one image in order to allow detailed study of the morphology of auroral forms. The analysis presented in this paper uses images taken by two different cameras stationed at KHO and Ny-Ålesund, approximately 120km northwest of KHO. Prior to 2015 both observatories were using CCD cameras that were equipped with narrow band-pass filters ($557.7\,\mathrm{nm}$, $630.0\,\mathrm{nm}$ and $427.8\,\mathrm{nm}$) and were part of the MIRACLE instrument suite (Sangalli et al., 2011). In 2015 a Sony $\alpha$7S full-frame all sky camera with a Sigma 8mm f/3.5 EX DG circular fisheye lens was deployed at KHO. The Sony camera has an exposure time of 4s and a variable time resolution of 12-30s.

ASC images are particularly useful in the effort to manually identify and qualitatively analyze auroral structures. However, manual identification and analysis of auroral forms is very time consuming and subject to human error and interpretation. For this reason, this study also uses the arciness index in order to quantify auroral morphology based on ASC images.

Furthermore, we report solar wind and IMF conditions as obtained by satellites in the OMNI-Network. They are located at the L1 point and their data are propagated to the magnetopause. The OMNI data have a temporal resolution of one minute.

## 3 Methodology

The initial identification of PMAFs was accomplished by manual inspection of keograms from the MSP at KHO. PMAFs manifest themselves as distinct structures in the red (630.0 nm) and green (557.7 nm) channels of a keogram. Their poleward motion causes them to appear in a keogram as a diagonally aligned band that fades over time. This is shown in Fig. 1 for two example events in the green and red channels of a keogram.

During the identification of PMAFs, several criteria would have to be met in order for a candidate event to be considered a PMAF event:

- Auroral emission in the 630.0 nm and 557.7 nm channels of the meridian scanning photometer at intensities significantly greater than typical noise levels ($\sim 100 - 200$ counts in Fig. 1).

- Poleward motion starting from the auroral oval

- Clear spatial separation between the PMAF candidate and other auroral emissions near the end of the candidate's lifetime.

- Occurrence within a time window of three hours before to three hours after magnetic noon (9-15 MLT).

For each PMAF event, start and end times were recorded (see Fig. 1). The start time was defined to be initial brightening of the auroral oval while the end time was defined by the fading of emissions to close to noise levels.

This paper presents analysis of two separate PMAF event lists, the first event list consisting of 23 PMAFs that occurred between 2003 and 2008. The second event list consists of 18 PMAF events that all occurred on the 18th of December 2017. It is unusual for that many PMAFs to be observed on a single day. This unusually high PMAF occurrence rate will be commented on in the discussion.

In regard to the qualitative description of PMAF morphology, both event lists are combined and analyzed jointly. In the quantitative analysis involving the arciness index, the event lists are kept separate due to differences in the instrumentation used.

The arciness index $A$ is a quantitative measure of how much the shape of auroral structures in an image resembles an arc. It was first introduced by Partamies et al. (2014) and proved to be effective at identifying auroral arcs in the dusk, night





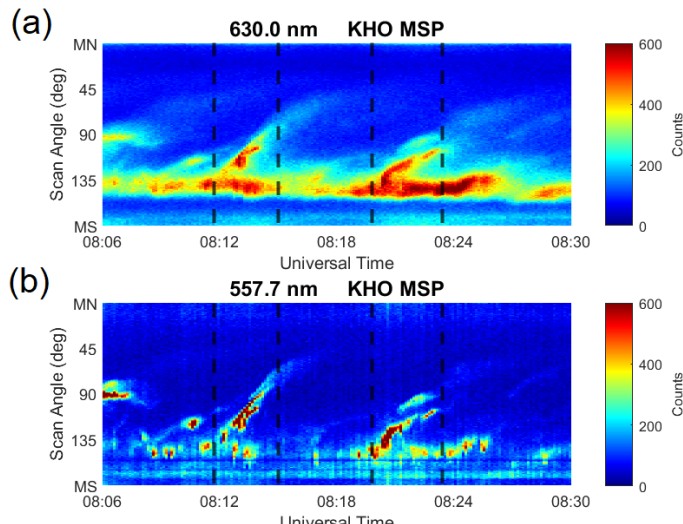

**Figure 1.** Uncalibrated keograms as a function of scan angle along the magnetic meridian from $0°$ (magnetic north, MN) to $180°$ (magnetic south, MS) in the red (a) and green (b) auroral emission lines showing two PMAF events as well as their start and end times delimited by black dashed lines. Time is displayed on the x-axis as universal time, which lags behind magnetic local time by approximately three hours in Svalbard.

and dawn sectors. This paper investigates the ability of the arciness index to identify arcs (PMAFs) in the noon sector ($9 - 15$ MLT). The index is a number between 0 and 1 and calculated by an algorithm for a gray-scaled ASC image. In the first event list the arciness algorithm uses gray-scaled images taken by MIRACLE ASCs at KHO and in Ny-Ålesund that had a narrow-band pass filter for the green auroral emission line mounted onto them. The arciness that was calculated in this manner will be referred to as narrow-band arciness.

This is different from the way the arciness of the second event list is determined. In the second event list the arciness algorithm is fed gray-scaled versions of the green/red component of RGB images that were taken by the Sony camera without a narrow-band pass filter. This will be referred to as wide-band arciness.

In general, the algorithm identifies all structures comprised of the brightest pixels (referred to from here on as bright pixels) in a given image and fits each structure, defined by a clustering of bright pixels, with a polynomial. The index is defined as:

$$A = \min\left[\frac{3}{\ln(NM)}, 1\right] \tag{1}$$

where $N$ is the weighted number of dominant auroral structures in an image:

$$N = \left[\sum\left(\frac{n}{\sum n}\right)^c\right]^{-1/c} \tag{2}$$

with $c$ being equal to the number of structures and the sums summing over each structure $n$. $M$ is a measure of the typical deviation of the actual shape of the auroral structure from the fit. $M$ is given by the sum of $\chi^2$-values (goodness of fit parameter) of different auroral structures normalized to the number of pixels in all structures:

$$M = \frac{\sum \chi^2}{\sum n} \tag{3}$$

Due to the fact that $M$ is a measure of deviation from a fit, wider auroral structures have higher values of $M$ and therefore lower arciness values than thinner structures. Consequently, auroral structures whose shape can not be neatly described by a



| Event List | $N_{\text{PMAFs}}$ | $T_{\text{PMAF}}$ | ASC images | $B_y$ (nT) | $B_z$ (nT) | $V_{\text{SW}}$ (km/s) | $T_{\text{SW}}$ ($10^5$K) | $P_{\text{SW}}$ (nPa) |
|---|---|---|---|---|---|---|---|---|
| 1 | 23 | 7 min | narrow-band | 0.0 | -0.8 | 614 | 2.29 | 3.44 |
| 2 | 18 | 5 min | wide-band | 6.6 | -2.0 | 598 | 2.23 | 3.91 |

**Table 1.** Overview of both event lists showing the number of PMAFs $N_{\text{PMAFs}}$, the average PMAF lifetime $\bar{T}_{\text{PMAF}}$, the presence of a narrow-band filter on the ASC, the average IMF components $\bar{B}_y$ and $\bar{B}_z$ and average values for solar wind speed $\bar{V}_{\text{SW}}$, temperature $\bar{T}_{\text{SW}}$ and dynamic pressure $\bar{P}_{\text{SW}}$.

polynomial (e.g. diffuse aurora) have low values of $M$.

Since $N$ describes the weighted number of structures over which the brightest pixels are distributed, having most bright pixels concentrated in a single auroral structure (low values of $N$) leads to high arciness. This can be the case for an auroral arc. Conversely, having multiple auroral shapes with similar brightness in the same image causes the arciness to decrease.

Each structure is fitted separately with a polynomial. The order of the polynomial $p$ depends on the number of pixels $n$ in the auroral structure that is to be fitted $p = \log_{10}(n)$. Typically, PMAFs are fitted with parabolas and third order polynomials. For more detailed information on the algorithm determining arciness we refer the reader to Partamies et al. (2014).

It is interesting to note that due to arciness being defined as the minimum (eq. 1), there is an upper bound for arciness. This upper bound is $A = 1$, and can be reached often, up to more than half of the time in the dusk sector (Partamies et al., 2014). In this dataset (9-15 MLT) arciness reaches its upper bound less than 25% of the time.

Arciness has been used to investigate auroral forms occurring between 16 - 10 UT, and has recently been investigated during all local times (Partamies et al., 2022). However, this is the first time arciness is used to study morphology of dayside auroral features such as PMAFs.

For this study, Superposed epoch analyses (SEAs) were employed to investigate the evolution of arciness during the occurrence of PMAFs. As not all PMAF events are equal in duration (lifetime), they are normalized in order to enable comparison using SEAs. In this analysis, not only the time span in which a PMAF occurred is considered, but also one PMAF lifetime before the beginning and one lifetime after the end of an event.

Depending on the exact lifetime of each individual PMAF we used 1-2 minute time bins of the arciness data for the SEA. This corresponds to 3-4 time bins per event lifetime with about 5-10 data points per bin per PMAF.

In an SEA, as the number of intervals that one event lifespan is divided up into increases, the time resolution increases, but statistical fluctuations increase as well, as fewer data points are averaged in each interval. Hence, deciding the number of intervals that the events are divided into is a trade-off between time resolution and the minimization of statistical errors. In this study there are usually three or four time intervals per PMAF lifetime, which corresponds to a time resolution of approximately 1-2 minutes, depending on the exact lifetime of any given PMAF.

## 4 Results and Discussion

Table 1 gives an overview of the solar wind driving conditions as well as other pertinent information during both event lists. The mean value of IMF $B_z$ was negative in both datasets, as is to be expected (Drury et al., 2003; Fasel, 1995; Wang et al., 2016; Xing et al., 2012, 2013). Solar wind dynamic pressure $\bar{P}_{\text{SW}}$ and speed $\bar{V}_{\text{SW}}$ were slightly above their average respective values. This has been observed in the literature as well (Fasel, 1995).

### 4.1 Visual inspection of PMAFs

The analysis of ASC images from both event lists has yielded information regarding the general morphological evolution of PMAFs. This general evolution throughout the lifetime of a PMAF is as follows:

1. Just before the onset of a PMAF event the auroral oval rapidly expands equatorward and may subsequently retreat poleward. The event begins with auroral oval emissions in the $557.7\,\text{nm}$ (and to a lesser degree in the $630.0\,\text{nm}$ emission line) intensifying at the open-closed boundary (OCB) or slightly poleward of it, depending on IMF geometry and the magnetic local time of occurrence. For duskward IMF ($B_y > 0$) PMAFs are associated with open-closed boundary intensifications (OCBIs, commonly referred to as EBIs, however the authors believe the term OCBI is more appropriate) in the pre-noon



**Figure 2.** A series of ASC images showing the evolution of the morphology of a PMAF (white ellipse) in event list 1. The images were taken without a narrow-band filter at the Kjell Henriksen Observatory. The orientation of the images is magnetic N-W-S-E, starting from the up direction and going clockwise (see compass rose in image (a)). Image (a) was taken on 18 December 2017 at 06:15:13 UTC. Each following image was taken with a constant cadence of approximately 23 seconds.

sector and intensifications slightly poleward of the OCB in the post-noon sector, respectively. The opposite is true for dawnward IMF ($B_y < 0$) conditions.

2. Within the auroral oval an east-west aligned arc-like structure of enhanced auroral intensity appears (predominantly in the $557.7\,\mathrm{nm}$ emission line). This structure can appear all at once or it can form as a consequence of the merging of multiple spatially separated patches of aurora into an arc-like structure.

3. The PMAF separates from the auroral oval as it propagates poleward. There is also an azimuthal component in the motion of the PMAF, which is dependent on IMF $B_y$. Under eastward/duskward (westward/dawnward) IMF PMAFs move northwest (northeast). The orientation of the PMAF seems to be aligned azimuthally independent of IMF orientation.





**Figure 3.** Two series of ASC images showing large scale auroral patches marked by white ellipses merging into arc–shaped PMAFs. The images were taken in the same manner and have the same orientation as in Fig. 2. The cadence of this image series is approximately 12 seconds, with image (a) being taken at 09:24:33 UTC and image (e) being taken at 07:18:20 UTC on 18 December 2017.

4. As the PMAF is moving poleward it can expand and brighten azimuthally. The direction in which PMAFs expand and brightness spreads through the PMAFs shows the same dependence on IMF geometry as the direction of azimuthal motion of PMAFs: preferentially westward under duskward IMF ($B_y > 0$) and eastward under dawnward IMF ($B_y < 0$).

5. Nearing the end of the PMAF lifetime the PMAF starts to dim in the red and green emission lines. At this point, there may be re-brightening events in the PMAF where auroral intensities especially in the green emission line are enhanced. At approximately the same time there would be associated increases in auroral oval brightness.

6. Finally, the PMAF emissions start fading in all wavelengths and the PMAF may expand along the north-south dimension before it has faded completely.





Fig. 2 illustrates this evolution in a series of ASC images with a concrete example. The PMAF shown in the images occurred under IMF $B_y > 0$ on December 18th, 2017 at 06:15 UT, which corresponds to approximately 09:15 MLT. In image b), one can observe a somewhat weakly pronounced OCBI (blue arrow) followed by the appearance of an auroral structure defined by enhanced 557.7 nm emissions growing from east to west in images b) and c) (white ellipse). Most of the OCBI has occurred

before the onset of the PMAF and is not shown in Fig. 2. The PMAF develops into an arc-like structure aligned along magnetic east-west propagating towards northwest (images d)-g)). The brightness of the PMAF decreases from image h) to image j) before the PMAF brightens again in images k) and l). At approximately the same time, the auroral oval increases in brightness too (images j) to l)). Finally, auroral emissions associated with the PMAF fade completely in images m)-p).

An intensification at the OCB is expected to occur considering this event occurred in the pre-noon sector under positive IMF

$B_y$. As previously mentioned, the authors believe it is more appropriate to refer to this phenomenon as OCBI as opposed to EBI, since there can be closed field line precipitation leading to emissions equatorward of where the PMAF forms that are considered part of the auroral oval (see Fig. 2). Those emissions would occur equatorward of the OCB, however the intensification of auroral emission associated with the formation of a PMAF occurs at the OCB rather than the equatorward boundary of the auroral oval. Hence, it is more appropriate to refer to this phenomenon as OCBI.

The direction in which the PMAF grows in the early stages of its evolution is in agreement with the above description: west under duskward IMF. The same is true for the direction of propagation, which is towards northwest under duskward IMF.

The morphological evolution of the PMAF event in Fig. 2 does not include the merging of different auroral patches into an arc-like shape (see step 2), but rather the appearance of a brightening arc-like structure that is spatially connected at all times. Fig. 3 shows two examples of different auroral forms merging to form a PMAF. In the first example in Fig. 3 (images a)-d)),

a westward expanding arc consisting of strong 557.7 nm emission can be observed close to the southeastern horizon. During the expansion of this arc, another arc-like structure forms stretching from the western horizon to the southern sky. Both auroral forms expand eastward and westward until they eventually merge and form a PMAF in image d).

The second example in Fig. 3 includes multiple merging events. In images e) and f), a northwest-ward propagating PMAF can be observed that is brightening. The auroral form has moved further west in image g) and h) when a patch of 557.7 nm

emission appears eastward of the auroral form. This patch quickly expands westward forming another arc-like structure that proceeds to merge with the original auroral form to its west. The remaining images show a similar process occurring again. In images k) and l) another patch of intense 557.7 nm aurora appears eastward of the PMAF which subsequently goes on to expand westward to ultimately merge with the PMAF too.

The merging of auroral patches into a singular structure is interpreted in two different ways depending on the scale of the

30 auroral patches. Patches on the order of tens of kilometers separated by similarly sized regions devoid of 557.7 nm aurora may be the ionospheric manifestation of inhomogeneities in the spatial distribution of solar wind particles.

On larger scales, however, auroral structures on the order of ∼100 kilometers connecting to form a PMAF (as shown in Fig. 3) might indicate localized X-line reconnection occurring at different reconnection efficiencies at different locations. The phenomenon of localized dayside reconnection has been reported before. Maynard et al. (2006) have reported magnetic re-

35 connection occurring asynchronously leading to reconnection signatures being observed at separate regions in the ionosphere. Additionally, Sandholt et al. (2003) have reported activation of magnetopause reconnection at different locations based on data from low Earth orbit satellites.

Whether or not re-brightening events can be observed depends on the PMAF class as introduced by Fasel et al. (1994). Ap-

40 proximately 70% of PMAFs in this study fall into the PMAF2 category, with the remaining 30% being considered members of the PMAF1 class. While this class distribution is similar to the findings reported by Fasel (1995) the PMAF1 occurrence rate in our dataset is approximately twice as large compared the work by Fasel.

We observed several qualitative differences between class 1 and 2 PMAFs beyond the re-brightening characteristics that they are defined by. One of these differences concerns the lifetimes of events in both classes. We found PMAF2 events to last ap-

45 proximately 1 minute longer on average than PMAF1 events.

Considering that this classification scheme is based on the occurrence of re-brightening events in any given PMAF, it is intuitive that both classes of PMAFs would have different ranges of lifetimes. Fasel et al. (1992) explained the occurrence of re-brightening events in terms of multiple X-line reconnections along the same flux tubes. Each reconnection event would lead to a brightening of PMAFs due to Alfvén waves being launched from the reconnection site. This would provide class 2 PMAFs

with more solar wind plasma at later stages of the event time frame and thereby increase their lifetime.

Additionally, we observed generally lower levels of auroral brightness in the later stages of PMAF1 events than in PMAF2 events, which can also be understood in terms of the formation mechanism of PMAF2s presented by Fasel et al. (1992). This observation was also made by Sandholt and Farrugia (2007), who reported higher levels of 557.7 nm emissions at high latitudes in PMAF2s.

Moreover, Sandholt and Farrugia (2007) reported observation of a clear dependence of PMAF class on magnetic local time.



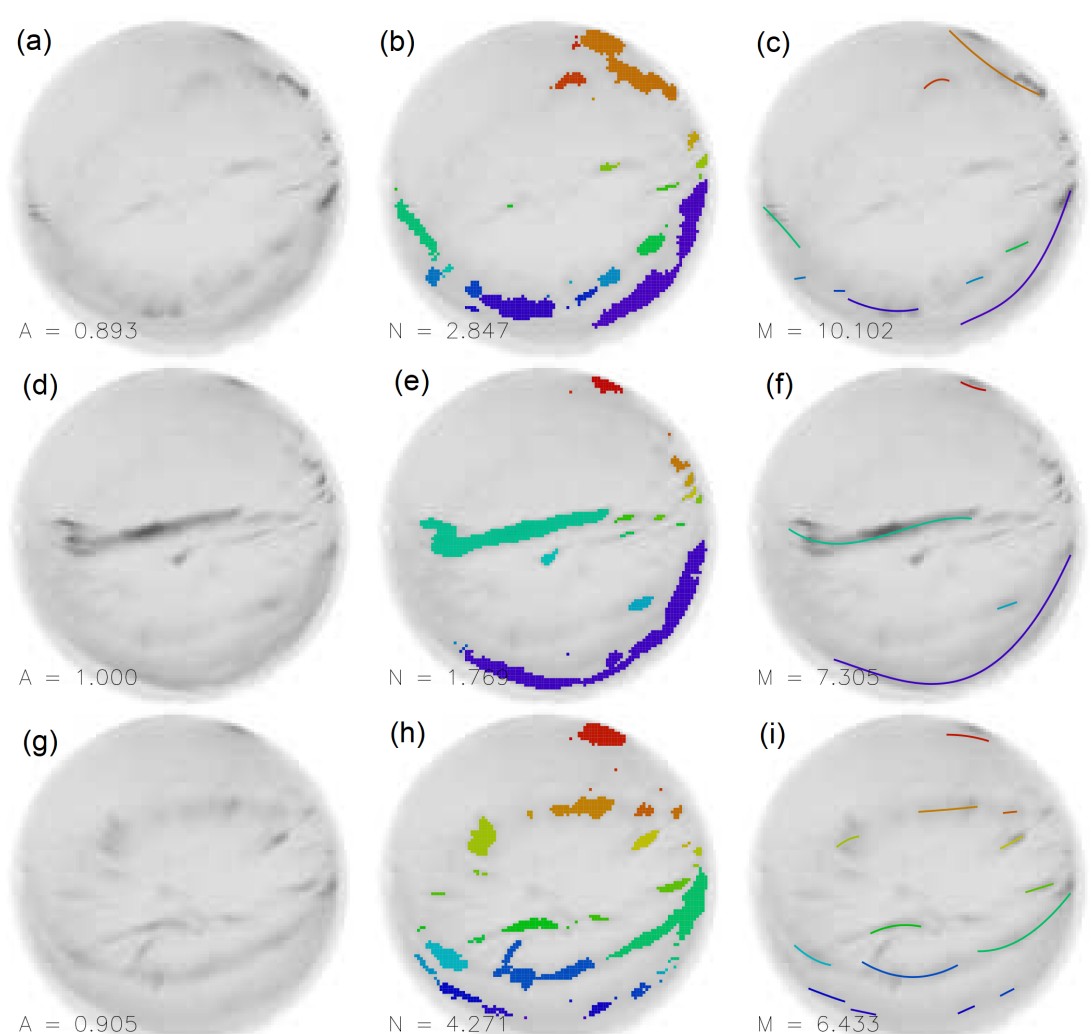

**Figure 4.** ASC Images a), h) and m) from Fig. 2 in an inverted gray scale with the associated arciness value $A$ (left column), weighted number of bright-pixel-clusters $N$ overlaid on original images using color-coding to distinguish between different clusters (center column), fit polynomials drawn onto original image for each cluster along with the $M$ value of each image (right column).

According to them, under IMF $B_y>0$ pre-noon events would fall into the PMAF2 class, while post-noon events would be exclusively PMAF1s. This situation would be reversed under IMF $B_y<0$. However, we have found no relationship between PMAF occurrence time and class. It has to be mentioned, that this study includes significantly more pre-noon events (32) than post-noon events (9). This is due to the second event list predominantly covering the pre-noon sector within a period from 09:00-13:00 MLT.

It is also worth mentioning that the second event list contains 18 PMAF events that all occurred in a single 4 hour time period. It is quite unusual to observe this many PMAFs in such a short time frame. This unusually high PMAF occurrence rate on this particular day has been addressed by Hwang et al. (2020) and is likely due to long lasting pulsed reconnection partly facilitated by steady IMF $B_y$ conditions (Hwang et al., 2020; Moen et al., 2012).





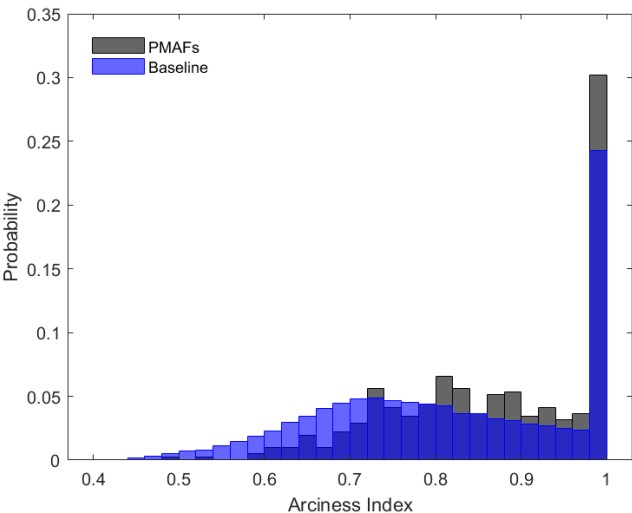

**Figure 5.** Arciness values during PMAF events (black) and daytime baseline (blue). The baseline is determined by taking all arciness values from the noon sector (9-15 MLT) of the days in the first event list, excluding times during which PMAFs occur.

## 4.2  Arciness Evolution during PMAFs

We have investigated how the arciness index is related to PMAF activity in order to use an objective and quantifiable measure of the morphology of PMAFs. We compared PMAF arciness values with arciness values between 9 MLT and 15 MLT outside of PMAF occurrence.

Fig. 4 shows three different ASC images and their associated arciness values (left column), $N$-values (center column) and $M$-values (right column). The images are of the same PMAF event as shown in Fig. 2 and correspond to the beginning of the event (top row, panel a) in Fig. 2), during the event (middle row, panel h) in Fig. 2) and near the end (bottom row, panel m) in Fig. 2). During this event arciness increases from the first image to the second one and has decreased again in the third image. PMAFs are expected to coincide with higher values of arciness, as they are bright arc-like structures, which would be well approximated by a polynomial. Additionally, auroral emissions associated with PMAFs are primarily green, while other dayside aurora is mainly red (Lorentzen et al., 2010). Since arciness (in the first event list) is calculated from images taken with a green narrow-band filter, this will lead to higher values of arciness too, as the brightest pixels are overwhelmingly concentrated in the PMAF. If a red filter had been used, other types of aurora that show up in the images would be home to many of the brightest pixels, leading to lower arciness.

Fig. 5 shows normalized histograms of arciness of PMAFs in the first event list (red) and of their baseline (blue). This baseline is comprised of all arciness values between 9 and 15 MLT on all days in the event list, except the times during which a PMAF occurs.

As seen in Fig. 5, PMAF arciness tends to be higher than general dayside arciness, however, this difference is quite small. The mean PMAF arciness is approximately $\bar{A}_{\mathrm{PMAF}} = 0.86$ while the mean baseline value is $\bar{A}_{\mathrm{baseline}} = 0.81$, a difference of roughly $0.4\sigma$ (The standard deviation of both distributions is $\sigma = 0.14$). This means, there is great overlap between typical arciness values during PMAF events and the baseline. Therefore, it is not possible to identify a PMAF merely based on the value of the arciness index alone.

A somewhat more complex approach considers the evolution of the arciness index over the course of a PMAF event. This approach is based on the assumption that even though the arciness index might not have particularly different values during PMAF events from the baseline, there might be a common temporal evolution arciness goes through in each event. In order to determine whether this is the case, an SEA was employed to analyze the behavior of narrow-band arciness during the 23 PMAFs in the first event list. The result is shown in panel (a) of Fig. 6. It shows the evolution of the arciness as a function of the PMAF age, starting from one PMAF lifetime before the beginning of the event to one lifetime after its end. The event time





Goertz et al.: Morphology of PMAFs                                                                                11

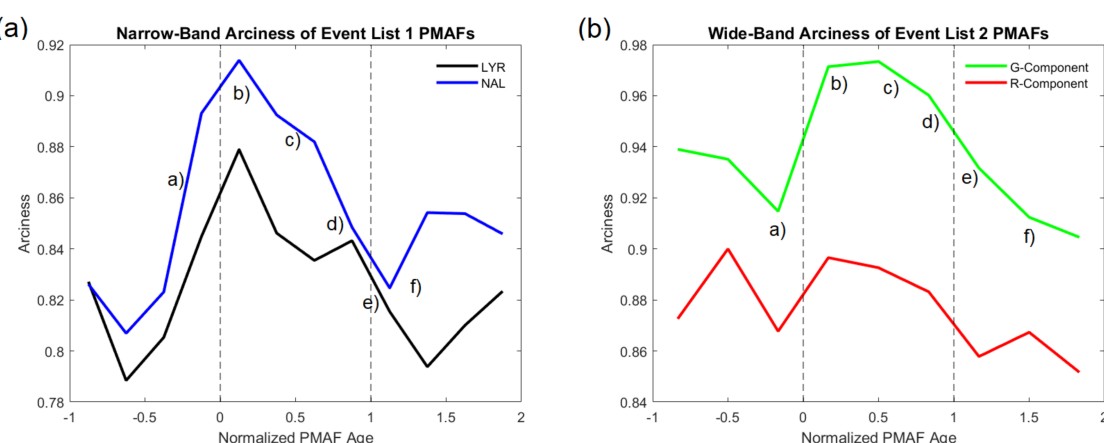

**Figure 6.** Superposed epoch analysis of arciness for several PMAF events. Left: narrow-band arciness calculated from ASC images taken in Ny-Ålesund/NAL (red) and Longyearbyen/LYR (blue). Right: Green and red wide-band arciness from KHO ASC images.

frame is delimited by dashed lines on both sides. Arciness has been calculated and analyzed for ASC images taken from both KHO/Longyearbyen (black) and Ny-Ålesund (blue). The arciness curves for the KHO and Ny-Ålesund data show a similar evolution: arciness increases approximately half a PMAF lifetime before the start of the event, peaking just after the start time, and subsequently decaying towards the baseline value near the end of the PMAF life.

Yet, there is one slight difference in the two curves. Ny-Ålesund arciness appears to have higher values generally. Since Ny-Ålesund is approximately 120 km north of the KHO, the ASC has a slightly different field of view. This implies that the auroral oval, which a given PMAF separates from near the start time of the event, will be closer to the horizon. Hence, fewer of the brightest pixels will be part of the auroral oval and more pixels will be concentrated in the PMAF itself, leading to higher arciness.

In an effort to reproduce the observed evolution of the arciness index over the course of a PMAF event, the same analysis was conducted with the second event list, which includes 18 events that occurred on a single day. On this day, however, filtered green emission images were not available from KHO. Instead, the arciness algorithm is fed the green or red component of the RGB color images. As previously mentioned, the arciness values calculated in this manner are referred to as wide-band arciness, while the original kind (ASC equipped with a narrow-band pass filter) are referred to as narrow-band arciness in this paper. The key differences between the ASC images taken with and without narrow band filters, are firstly, concerning the signal, that the green component of the color image contains aurora of the green emission line with contributions from the red emission line, while images taken with a green narrow-band filter only include green aurora. Secondly, concerning the background, filtered ASC images only contain a small amount of background radiation coming from the narrow range of wavelengths specified by the filter, while any given color-component of non-filter images will include background radiation from all wavelengths and from any source, such as scattered sun-/moon light, light pollution etc. This implies, that, unsurprisingly, filtered images have significantly better signal to noise ratios compared to their unfiltered counterparts, which means the peak in narrow-band arciness associated with PMAF occurrence is expected to be greater.

Panel (b) in Fig. 6 shows the result of an SEA applied to green and red wide-band arciness for event list two PMAFs. The general evolution of green wide-band and narrow-band arciness seem to be almost identical: increasing arciness towards the beginning of the event, a peak during the event, and a decay towards the end. In principle, the red wide-band arciness follows the same curve. The increase in arciness is less pronounced though, which has to do with the relative emission strength of the auroral oval and the PMAF. In the green emission line, the PMAF is much brighter than the auroral oval, whereas they are similarly bright in the red channel (see Fig. 1). Additionally, due to their higher emission lifetime, red emissions associated with the PMAF are more diffuse, leading to a broader spatial distribution of bright pixels in the ASC image. Consequently, red wide-band arciness does not rise by the same amount.

In order to understand the behavior of the arciness index in relation to PMAF events generally (see Fig. 4 and Fig. 6), it is highly instructive to revisit the typical evolution of PMAFs in ASC images. Fig. 7 shows six ASC images taken with a



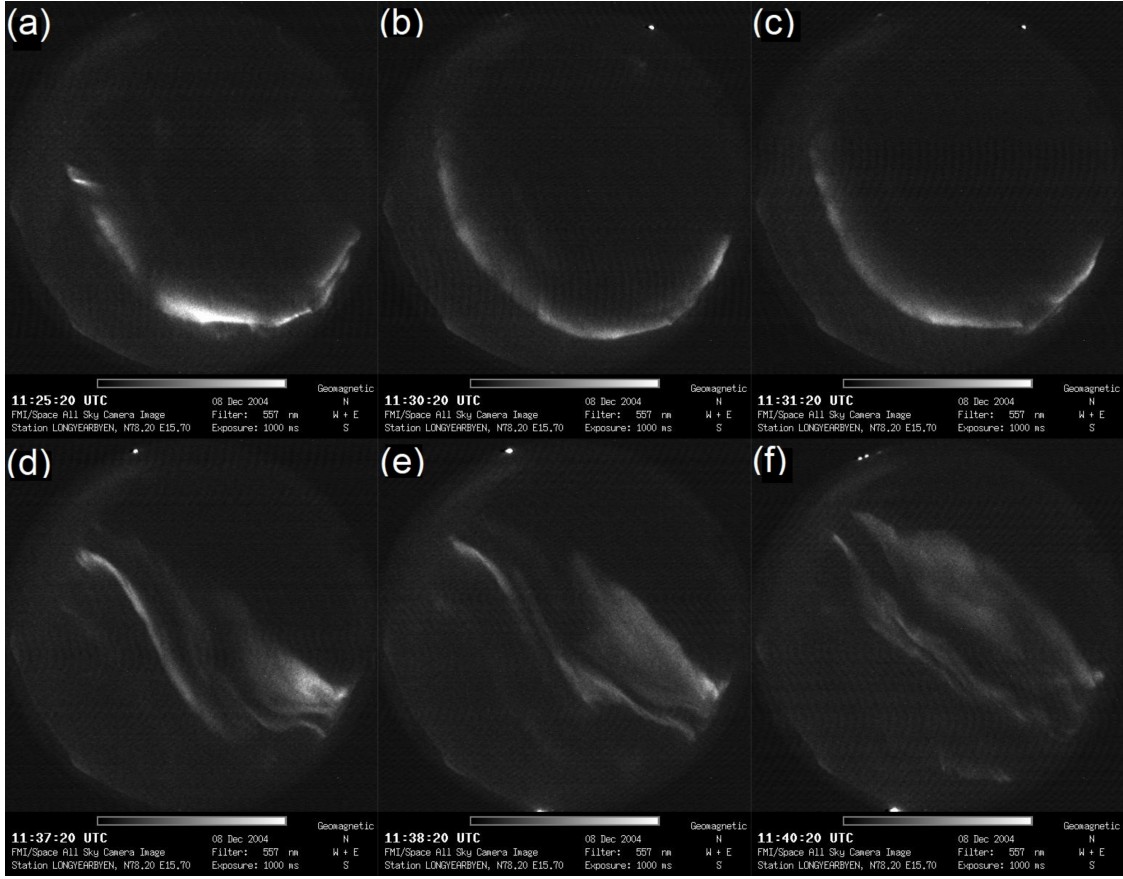

**Figure 7.** Several ASC images mounted with a narrow-band filter ($\lambda = 557.7\,\mathrm{nm}$) portraying the typical evolution of a PMAF. Compared to Fig. 2 and Fig. 3, the orientation of the images here is flipped (see compass rose).

narrow-band pass filter ($\lambda = 557.7\,\mathrm{nm}$) of the same PMAF in different stages of its development similar to Fig. 2. Image (a) shows the bright auroral fragments aligned along an east-west stretching band just before the birth of the PMAF. The different fragments merge into a singular arc in image (b), which begins to move poleward (c). The merging of multiple fragments into a singular arc significantly reduces the number of structures bright pixels are distributed across. Generally, although not clearly

visible in Fig. 7, the OCBI increases the brightness in the auroral oval, causing a higher percentage of the brightest pixels to be in the auroral oval. Those two effects together, the combining of multiple fragments into an arc, and the brightening associated with the OCBI, lead to a sudden increase in arciness around the beginning of the PMAF lifetime.

As the PMAF fully separates from the oval, the arciness starts to decrease again, as the separation increases the number of structures in the image. During the poleward motion of the PMAF the width of the arc along the north-south dimension

increases and it can split into multiple arcs of varying brightness (d). Both of these factors generally cause a reduction in arciness. Near the end of the event the PMAF begins to lose its arc-like shape and spreads out to gradually become more diffuse, causing a final drop in arciness around the end time (e). Finally, auroral emissions in all wavelengths fade as the PMAF event ends (f) and the arciness returns to its baseline. It should be mentioned that this is not (merely) a description of arciness during the event in Fig. 7, but rather of the general evolution of arciness relating to PMAF occurrence.

The automation of PMAF detection has been found to be very difficult. In the future our findings might enable the development of an automated PMAF detection algorithm based on the arciness index. This algorithm may search arciness data for the typical evolution in relation to PMAF occurrence that is presented in Fig. 6. According to our study, this would be possible



with wide- and narrow-band arciness.

## 5    Conclusions

In this paper we present a qualitative description and discussion of the morphological evolution of PMAFs using optical anal-    5
ysis based on ASCs. We break this evolution down into six steps, including (1) an equatorward expansion and intensification
of the open-closed boundary, (2) the appearance of a complete arc-like structure or auroral patches merging into an arc-like
structure, (3) poleward and westward (eastward) propagation under IMF $B_y$>0 ($B_y$<0), (4) expansion or brightening spreading
through the PMAF in the direction of azimuthal propagation, (5) possible re-brightening events coinciding with auroral oval
intensifications and finally (6) dimming and expansion along the north-south meridian as the PMAF fades away.    10
While there have been previous reports of PMAF morphology, this paper presents the most detailed description published
to date including novel aspects. One such aspect is the merging of auroral patches/arcs into larger scale PMAFs, which we
suggest might be explained by localized dayside magnetopause reconnection.
Furthermore, we have found differences between PMAF1 and PMAF2 events beyond their defining re-brightening charac-
teristics. In this dataset, PMAF2 events have stronger $557.7\,\mathrm{nm}$ emissions than PMAF1 events at higher latitudes as well    15
as lifetimes that are approximately 1 minute longer. Both of these effects could be explained by the re-brightening events
exhibited by class 2 PMAFs being manifestations of multiple X-line reconnection events on the same flux tubes as has been
proposed previously.
Unlike previous reports on this matter, we have found no relationship between PMAF class and occurrence time.    20

In the quantitative part of the study, the behavior of the arciness index in relation to PMAF activity is investigated.
While arciness values during PMAF activity are generally higher than during the rest of the dayside, this effect is relatively
minor, as there is much more overlap than difference. Yet, a superposed epoch analysis uncovered a characteristic evolution
the arciness value exhibits over the course of a PMAF event, in which arciness increases due to the formation of a PMAF
and subsequently decays back to its original value as the PMAF propagates poleward and fades over time. Each part of the    25
characteristic evolution arciness goes through can be understood based on morphological changes of the PMAF as well as the
auroral oval. This relationship between arciness and PMAF occurrence has been found to hold true regardless of variations
in the input to the arciness algorithm and the manner in which ASC images are taken. These findings pave the way for the
development of an automated PMAF detection algorithm based on arciness. Automatic detection of auroral forms based
on optical data has been found to be extremely difficult, which makes the possibility of an arciness-based PMAF detection    30
algorithm very attractive.

#### Data Availability

Solar wind and IMF data can be retrieved from the OMNIWeb database (https://omniweb.gsfc.nasa.gov, last access 25 April
2022). MIRACLE ASC quicklook data are available at (https://space.fmi.fi/MIRACLE/ASC/, last access 25 April 2022), full
resolution image data can be requested from FMI (kirsti.kauristie@fmi.fi). Quicklook Sony ASC images are available on the    35
KHO website (http://kho.unis.no, last access 25 April 2022).

#### Author Contributions

All authors have contributed to the discussion of the results and the writing of the paper.

    40
#### Competing Interests

The authors declare no conflicts of interest.

#### Disclaimer

This study is based on parts of AG's Master thesis "Poleward moving auroral forms and dayside flow channels", which is    45
available on the KHO website: https://kho.unis.no/doc/MSc_Goertz.pdf, last access 25 April 2022.

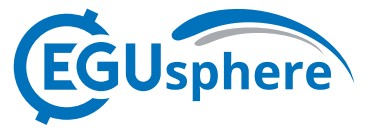

**Acknowledgments**

The work by NP & LB is supported by the Norwegian Research Council (NRC) under CoE contract 223252. DW was supported by a NERC Independent Research Fellowship (NE/S015167/1). AG thanks René Reifarth for making his Master's project possible, which led to this publication. The authors thank the KHO team and PI Dag Lorentzen for maintenance of the auroral color camera and MSP. We further thank S. Mäkinen, J. Mattanen, A. Ketola, and C.-F. Enell for maintaining MIRACLE camera network and data flow. The NAL all-sky camera is funded by the PNRA and the INAF-IAPS, its operation is also supported by the staff of the Dirigibile Italia Station and the Institute of Polar Sciences of CNR.

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
