# Peer review of "Morphological Evolution and Spatial Profile Changes of Poleward Moving Auroral Forms"

_EGUsphere, 2022_

## Referee Comment (RC2)

Reviewer report of "The Morphology of Poleward Moving Auroral Forms" by Anton Goertz, Noora Partamies, Daniel Whiter, and Lisa Baddeley

In this study, the authors observed carefully the spatial and temporal morphological profiles of poleward moving auroral forms (PMAFs) based on the data obtained from the scanning photometer and all-sky camera image data and found new feature on PMAFs; the merging of auroral patches into a singular arc-like structure, which can be considered that auroral structures with the order of  $\sim$  100 kilometers can be led to form a PMAF within a large-scale physics. This merging might be addressed by the localized dayside magnetopause reconnection.

This reviewer considers that some new features previously unrevealed can be explained by this study, but this paper does not yet reach the publication level in a present form because of following reasons.

**Major comments:**

1) On the use of terminology of "morphology";

This reviewer considers that this terminology is frequently used to point out "form" and "shape" of thing in our field. If you say, "the morphology of Poleward Moving Auroral Forms", readers may think that PMAFs have various kinds of shape or form. However, the authors discuss that PMAFs, which temporally and spatially changed, in this paper. The implication of "morphology" used in this study seems to be wrong. Then, this reviewer suggests replacing "morphology" with the other word, such as "temporal and spatial (profile) changes". In particular, this reviewer strongly felt that this section did state "temporal and spatial changes of auroral arcs (or PMAFs, but it actually remains question whether or not the whole process as shown in Figure 2 is "PMAFs"). The title should also be changed. This reviewer's candidate is "Temporal and Spatial (Profile) Changes of Poleward Moving Aurora Forms: Observations Based on All-sky Camera and Scanning Photometer (at Svalbard)".

2) The database compiling;

In this study, the authors promote the discussion on the PMAFs using two databases; one is the statistical database of PMAFs which were detected from 2003 to 2008, and another is the unusual and multiple PMAF events occurred on 18th December 2017.

The 1st database includes each individual PMAF event occurred in a day from 2003 to 2008? If only one PMAF event usually occurs in a day, the 2nd database should be identified as "anomalous" event. If so, can the authors discuss these two databased within the same work frame? The physics, such as formation mechanism, IMF conditions, and background magnetic field/plasma characteristics during the 2nd PMAF event might be different from the usual PMAFs (the 1st PMAF database)?

**3) The data structure;**

"Depending on the exact lifetime of each individual PMAF we used 1-2 minute time bins of the arciness data for the SEA. This corresponds to 3-4 time bins per event lifetime with about 5-10 data points per bin per PMAF." The relation between the time resolution of PMAF all-sky image data and arciness time bins is unclear and so complicated. Please explain more clearly with an illustration or rewrite this sentence more detailed.

**4) Dayside reconnection evidence;**

The authors assert through this study that PMAFs can closely be connected (linked) with dayside reconnection based on the previous studies. However, in this study, you do not show any clear observational evidence for the occurrence of dayside magnetic reconnection associated with PMAFs. The reconnection evidence can be obtained from in-situ space-based and remotely ground-based observations, such as the HF radar arrays (SuperDARN radars). At least, the authors should show some examples (data) of dayside reconnection evidence, if the PMAFs are associated with the dayside magnetospheric processes.

**5) Statistics of arciness;**

The tendency as shown in Figure 6 is varied depending on the IMF and solar wind conditions? Although the authors show the average profiles of IMF and solar wind plasma, actually, the PMAF events should occur under various solar wind conditions. If the authors try to examine statistical characteristics of arciness, the PMAF data under the specific or average IMF-By and -Bz and solar wind plasma conditions (as seen in Table 1) were used? Although the authors tell that "SEA was employed to analyze the behavior of narrow-band arciness during the 23 PMAFs in the first event list." and "the same analysis was conducted with the second event list, which includes 18 events that occurred on a single day", these PMAFs (23 events in first event list and 18 events in the second list) were occurring under the similar solar wind and IMF conditions or average solar wind conditions as shown in Table 1?

**Minor comments:**

1) In Abstract and everywhere: What is the definition of "open-closed boundary"? Is it the same region as the poleward edge of the main aurora oval?

2) Table 1; Why don't the authors show the average value of plasma number density ( $N_p$ )? This reviewer considers that the solar wind density is more effective parameter in auroral phenomena than the solar wind temperature ( $T_{sw}$ ).

3) Section 4.1; Here, the authors tried to state the profile changes of PMAFs, but this reviewer feels just like reading several sentences as written in the research note. In these items, there are some PMAF signatures that have already well-known. The reviewer recommends re-organizing or re-structuring this section. In order to concisely and shortly show these series of spatial and

temporal PMAF change flows, how about illustrating these using the block diagrams?

4) "The merging of auroral patches into a singular structure is interpreted in two different ways depending on the scale of the auroral patches." "Patches on the order of tens of kilometers separated by similarly sized regions devoid of 557.7nm aurora may be the ionospheric manifestation of inhomogeneities in the spatial distribution of solar wind particles."

Can you provide the associated references? Or these are your considerations? If the latter case, why can you consider these?

5) What is "PMAF1 category" and "PMAF2 category"? What do PMAFs 1 and 2 have the significant characteristics?

6) What is the definition of "re-brightening events"? Please explain these phenomena more clearly.

7) Figures 2, 3 and 4; The explanations of these two figures are complicated. The author should show the time on the top of each panel, such as "(a) 6:15:13 UT (b) 6:15:35 UT...". In particular, in Figures 2 and 3, the title should be put. For example, "ASC images on 18th December 2017". In relation to this, this reviewer recommends that the authors should put a movie of ASC during the time intervals when you are discussing here (18th December 2017) as "supplementary information".

8) Figure 4; What is (are) the color code (colored regions and curves) assigned? The highness of arciness index? If so, please put a color bar to easily understand what color shows. Please explain more clearly how the colored region shown in the center column, and colored curves in the right column were calculated.

9) Figure 5; This reviewer cannot find the red part. Maybe you changed the color from red to black?

10) Figure 6; What do the labels from (a) to (f) seen in Figures 6 mean? This reviewer cannot find the explanations (notations) on these labels in the manuscript. Maybe, these labels are related to Figure 7?

11) In relation to 10) and Figure 7; Why do the authors need to independently show the PMAF's images here? Readers must become confused. If the authors want to discuss the arciness index variations associated with the PMAF's evolution, they should discuss this with a combination of Figure 6 with Figure 7. This reviewer recommends re-organizing these figures and associated sentences (paragraphs).

12) Section 5 (pp.13, L5 and L11); morphological evolution of PMAFs → temporal and spatial profile changes of PMAFs.
PMAF morphology → A series of temporal and spatial PMAF changes

13) Section 5 (pp.13, L12); which  $\rightarrow$  where or that

14) Section 4.2 (pp.12, L16) and Section 5 (pp.13, Ls 28 - 31); This reviewer considers that the automation of PMAF detection might has already been started to be developed based on machine learning technique (e.g., Convolutional Neural Network; CNN). Do you have any opinions on the event search using machine learning? If yes, you also should discuss the relation between your opinion and machine learning technique in the manuscript.

There are several recent reports that, with a help of machine learning technique, the auroras detected by the all-sky camera can automatically be categorized. The corresponding links are shown as follows.

<Reference>

1.https://www.nature.com/articles/s41598-022-11686-8

2.https://tromsoe-ai.cei.uec.ac.jp/#/

3.https://www.sciencedirect.com/science/article/pii/S1364682622000797?via%3Dihub#fig1

This reviewer considers that your research results and principle of the PMAF event search can be implemented to these algorithms. However, on the other hand, independently, will the authors build some system to automatically detect the PMAFs in near future?

---

## Author Comment (AC1)

**Author Response to RC2**

August 25, 2022

*Major comments:*
*1) On the use of terminology of "morphology";*
*This reviewer considers that this terminology is frequently used to point out "form" and "shape" of thing in our field. If you say, "the morphology of Poleward Moving Auroral Forms", readers may think that PMAFs have various kinds of shape or form. However, the authors discuss that PMAFs, which temporally and spatially changed, in this paper. The implication of "morphology" used in this study seems to be wrong. Then, this reviewer suggests replacing "morphology" with the other word, such as "temporal and spatial (profile) changes". In particular, this reviewer strongly felt that this section did state "temporal and spatial changes of auroral arcs (or PMAFs, but it actually remains question whether or not the whole process as shown in Figure 2 is "PMAFs"). The title should also be changed. This reviewer's candidate is "Temporal and Spatial (Profile) Changes of Poleward Moving Aurora Forms: Observations Based on All-sky Camera and Scanning Photometer (at Svalbard)".*

We agree that the term morphology is typically used to refer to the shape and form of auroral structures. However, we believe the term morphology is appropriate in describing the contents of this paper, as we do discuss the morphology (shape, form and brightness) of PMAFs both qualitatively and quantitatively using the arciness index. In fact, the arciness index is a quantifier for auroral morphology at a specific point in time, and does not describe changes in the morphology.
As you do point out in your minor comments, some instances of our use of the word morphology in this manuscript are inappropriate, and we will replace the term with 'spatial profile changes'.
Thank you for your suggestion of a new title. We do believe the phrase 'spatial profile changes' also accurately describes the content of this paper. For the title, we suggest 'Morphological evolution and spatial profile changes of poleward moving auroral forms'.

*2) The database compiling;*
*In this study, the authors promote the discussion on the PMAFs using two databases; one is the statistical database of PMAFs which were detected from 2003 to 2008, and another is the unusual and multiple PMAF events occurred on 18th December 2017.*
*The 1st database includes each individual PMAF event occurred in a day from 2003 to 2008? If only one PMAF event usually occurs in a day, the 2nd database should be identified as "anomalous" event. If so, can the authors discuss these two databased within the same work frame? The physics, such as formation mechanism, IMF conditions, and background magnetic field/plasma characteristics during the 2nd PMAF event might be different from the usual PMAFs (the 1st PMAF database)?*

The first database contains 23 PMAFs that occurred between 2003 and 2008. Those events took place on 10 different days, hence, some of those days had only 1 PMAF event, while on other days multiple PMAFs occurred. The second event list contains 18 PMAFs that all occurred on the same day. As you correctly point out, this is unusual and thus we separate the two databases and analyze them separately. Another reason we kept the two event lists separate is that the ASC images were taken with different setups (different cameras, and narrow-band filter present only in during event list 1). Consequently, the arciness algorithm was fed filtered 557.7 nm images in event list 1 and the

green (and red) component of the unfiltered RGB images in event list 2. We were unsure of how this qualitative difference in the ASC image data would manifest itself in arciness.

While the databases are different, we believe they can still be analyzed and discussed in the same study. As shown in table 1 in the manuscript, which summarizes the IMF and solar wind conditions during the two databases, the IMF and solar wind conditions are very similar for both event lists. Furthermore, both our manual inspection of ASC images as well as our quantitative analysis of the arciness index in relation to PMAFs have not revealed any qualitative differences between the PMAFs in the two event lists. We will ensure our reasoning behind the compiling of our event lists is clear in the manuscript.

*3) The data structure;*
*"Depending on the exact lifetime of each individual PMAF we used 1-2 minute time bins of the arciness data for the SEA. This corresponds to 3-4 time bins per event lifetime with about 5-10 data points per bin per PMAF." The relation between the time resolution of PMAF all-sky image data and arciness time bins is unclear and so complicated. Please explain more clearly with an illustration or rewrite this sentence more detailed.*

Thank you for bringing this to our attention. We agree this is hard to understand and believe this paragraph is easier to understand in the following form: *"Since PMAFs have different lifetimes, we normalize each arciness evolution curve associated with each PMAF. We then bin the arciness data into 3-4 time bins per PMAF lifetime. The first time bin corresponds to the mean of all arciness values in the first third (quarter) of all PMAF events. The second time bin corresponds to the mean of all arciness values in the second third (quarter) of all PMAF events, and so on. For PMAFs with a lifetime of 3 (4) minutes, this corresponds to a time resolution of 1 min. For PMAFs with a lifetime of 6 minutes however, this corresponds to a time resolution of 2 min."*

*4) Dayside reconnection evidence;*
*The authors assert through this study that PMAFs can closely be connected (linked) with dayside reconnection based on the previous studies. However, in this study, you do not show any clear observational evidence for the occurrence of dayside magnetic reconnection associated with PMAFs. The reconnection evidence can be obtained from in-situ space-based and remotely ground-based observations, such as the HF radar arrays (SuperDARN radars). At least, the authors should show some examples (data) of dayside reconnection evidence, if the PMAFs are associated with the dayside magnetospheric processes.*

While we understand the reviewers desire for evidence of dayside reconnection associated with the PMAF events in this study, we do not believe it is necessary to show data supporting reconnection taking place. This is because: while there is debate about whether or not *all* PMAFs are the ionospheric signatures of (pulsed) dayside reconnection, it is widely accepted that dayside reconnection is the most common (and perhaps only) driver of PMAF events. Furthermore, this study does not address the whether or not the PMAFs in our database are signatures of dayside reconnection/FTEs, nor does it attempt to. Our focus for this paper is the evolution of the morphology of PMAFs. We will edit the manuscript to explicitly state that we assume PMAFs are generally driven by pulsed dayside reconnection and that we do not attempt to address the cause of the PMAFs in our study.

*5) Statistics of arciness;*
*The tendency as shown in Figure 6 is varied depending on the IMF and solar wind conditions? Although the authors show the average profiles of IMF and solar wind plasma, actually, the PMAF events should occur under various solar wind conditions. If the authors try to examine statistical characteristics of arciness, the PMAF data under the specific or average IMF-By and -Bz and solar wind plasma conditions (as seen in Table 1) were used? Although the authors tell that "SEA was employed to analyze the behavior of narrow-band arciness during the 23 PMAFs in the first event list." and "the*

*same analysis was conducted with the second event list, which includes 18 events that occurred on a single day", these PMAFs (23 events in first event list and 18 events in the second list) were occurring under the similar solar wind and IMF conditions or average solar wind conditions as shown in Table 1?*

The solar wind and IMF parameter values listed in table 1 are averages for both event lists/databases. The difference between the two panels in fig. 6 is very unlikely to be related to differences in solar wind and IMF conditions during the two event lists, as they are quite similar (see table 1). The main difference between the data shown in the two panels of fig. 6 is the ASC image data. The first event list consists of ASC images that were taken by CCD cameras equipped with narrow band pass filters that were part of the MIRACLE instrument suite. In the second event list, the ASC images were taken by a SONY $\alpha7S$ camera and were not equipped with filters. Instead, the gray-scaled version of the green and red components of the RGB images were used to calculate arciness.
The analysis conducted in order to arrive at the data shown in panels a) and b) of fig. 6 are identical, the only difference lies in the image data and the event list.

*Minor comments:*
*1) In Abstract and everywhere: What is the definition of "open-closed boundary"? Is it the same region as the poleward edge of the main aurora oval?*

The OCB is the boundary between the ionospheric domains where magnetic field lines are closed (for example plasma sheet) and where they are open (for example the cusp region). This boundary corresponds to the poleward edge of the auroral oval on the nightside, and to the equatorward edge of the auroral oval on the dayside at noon. It is at the OCB where PMAFs first appear before they propagate poleward. However, there can be precipitation on closed field lines causing aurora on the dayside (close to dusk and dawn, further away from magnetic noon), which means the OCB is then poleward of the equatorward edge of the auroral oval. And hence, they are not the synonymous on the dayside.
In the literature, the term people typically use to refer to the place at which PMAFs first appear is the equatorward boundary of the auroral oval. This is slightly inaccurate, as PMAFs appear at the OCB. The equatorward boundary of the auroral oval and the OCB are the same close to magnetic noon, however they diverge closer to dusk/dawn.
We will ensure our reasoning behind the use of this terminology is clearly explained in the manuscript.

*2) Table 1; Why don't the authors show the average value of plasma number density (Np)? This reviewer considers that the solar wind density is more effective parameter in auroral phenomena than the solar wind temperature (Tsw).*

We are happy to replace the solar wind temperature value in table 1 with the average solar wind number density.

*3) Section 4.1; Here, the authors tried to state the profile changes of PMAFs, but this reviewer feels just like reading several sentences as written in the research note. In these items, there are some PMAF signatures that have already well-known. The reviewer recommends re-organizing or re-structuring this section. In order to concisely and shortly show these series of spatial and temporal PMAF change flows, how about illustrating these using the block diagrams?*

We decided to present our understanding of the morphological evolution of PMAFs in itemized form in order to concisely and efficiently convey our findings. It is true that some of the aspects mentioned in our items in section 4.1 are not novel features, but have already been reported in other publications. We nevertheless decided to include that information to present a full picture of the evolution of PMAFs, as well as to confirm previous studies that commented on this topic.
We will surely consider illustrating the itemized part of section 4.1, however we believe an illustration

could not replace the text, and both the text and the illustration would compliment each other.

*4) "The merging of auroral patches into a singular structure is interpreted in two different ways depending on the scale of the auroral patches." "Patches on the order of tens of kilometers separated by similarly sized regions devoid of 557.7nm aurora may be the ionospheric manifestation of inhomogeneities in the spatial distribution of solar wind particles."*
*Can you provide the associated references? Or these are your considerations? If the latter case, why can you consider these?*

These are our ideas for possible explanations for the phenomenon related to the merging of auroral patches into a PMAF. The very beginning of a PMAF event is marked by sudden increase in auroral brightness of the OCBI, caused by the arrival of electrons accelerated along magnetospheric field lines during dayside magnetopause reconnection. As the energy of electrons increases the travel time from the magnetopause to the ionosphere decreases. Hence, higher energy electrons arrive in the ionosphere sooner than their lower energy counterparts. Inhomogeneities in the spatial or energy distribution of solar wind electrons may manifest in the polar ionosphere as a patches of aurora appearing at slightly different times, depending on the level of electron flux and the electron energy. We will include these considerations in the manuscript to justify our suggestion for a possible explanation.

*5) What is "PMAF1 category" and "PMAF2 category"? What do PMAFs 1 and 2 have the significant characteristics?*

The PMAF categorization scheme was first introduced by Fasel (1994) according to the presence of re-brightening events and poleward propagation of PMAFs. This categorization scheme is as follows:
PMAF1: propagates poleward and fades
PMAF2: propagates poleward and re-brightens before fading
PMAF3: propagates poleward, re-brightens and slows down before fading.
The categorization is explained on page 2 in line 31 of the manuscript, and in the original 1994 paper by Fasel (DOI: 10.1007/978- 94-011-1052-5_15)

*6) What is the definition of "re-brightening events"? Please explain these phenomena more clearly.*

A re-brightening event refers to the sudden increase in auroral intensity of a PMAF after the PMAF has started propagating poleward. PMAF2s (and PMAF3s) are characterized by the occurrence of re-brightening events, while PMAF1s do not re-brighten. Frequently, the auroral oval equatorward of the PMAF brightens at approximately the same time. This is mentioned in the manuscript on page 8 line 6:
"The brightness of the PMAF decreases from image h) to image j) [fig. 3] before the PMAF brightens again in images k) and l). At approximately the same time, the auroral oval increases in brightness too (images j) to l)). Finally, auroral emissions associated with the PMAF fade completely in images m)-p)."

*7) Figures 2, 3 and 4; The explanations of these two figures are complicated. The author should show the time on the top of each panel, such as "(a) 6:15:13 UT (b) 6:15:35 UT...". In particular, in Figures 2 and 3, the title should be put. For example, "ASC images on 18th December 2017". In relation to this, this reviewer recommends that the authors should put a movie of ASC during the time intervals when you are discussing here (18th December 2017) as "supplementary information".*

Each ASC image in fig. 2 and fig. 3 will get a timestamp and a title header will be added to fig. 2 and fig. 3. We are happy to put together a .gif/video of the ASC image series and attach it as supplementary information.

*8) Figure 4; What is (are) the color code (colored regions and curves) assigned? The highness of arciness index? If so, please put a color bar to easily understand what color shows. Please explain more clearly how the colored region shown in the center column, and colored curves in the right column were calculated.*

The different colors in the middle column in fig. 4 corresponds to different structures. Each pixel belonging to a given auroral structure is highlighted in a different arbitrary color. The colored curves in the right column correspond to the fit polynomial of each auroral structure obtained from a least-squares fit. We will amend the caption of fig. 4 to make this clear in the manuscript.
We decided against describing the arciness algorithm in any more detail than we did in the section 3. Instead, we refer the reader to Partamies (2014) (DOI: 10.1002/2013JA019631, section 3.2), where the arciness algorithm is described in detail.

*9) Figure 5; This reviewer cannot find the red part. Maybe you changed the color from red to black?*

Thank you for catching this error, you are correct, we changed the color of the baseline histogram. The text has been corrected.

*10) Figure 6; What do the labels from (a) to (f) seen in Figures 6 mean? This reviewer cannot find the explanations (notations) on these labels in the manuscript. Maybe, these labels are related to Figure 7?*

Yes, the labels in fig. 6 correspond to approximate steps in the evolution of PMAFs and PMAF arciness, and they are explained in the second to last paragraph of section 4.2. And they are related to fig 7. We will add appropriate comments to the text and the caption of fig. 6.

*11) In relation to 10) and Figure 7; Why do the authors need to independently show the PMAF's images here? Readers must become confused. If the authors want to discuss the arciness index variations associated with the PMAF's evolution, they should discuss this with a combination of Figure 6 with Figure 7. This reviewer recommends re-organizing these figures and associated sentences (paragraphs).*

Yes, the reasoning behind showing PMAF images here is to give another example of the evolution of the spatial profile of PMAFs and then, more importantly, to explain the arciness evolution based on the morphology (shape, size and brightness) of the PMAFs. We will change that section of the text to make that more clear.

*12) Section 5 (pp.13, L5 and L11); morphological evolution of PMAFs --> temporal and spatial profile changes of PMAFs.*
*PMAF morphology --> A series of temporal and spatial PMAF changes*
*13) Section 5 (pp.13, L12); which --> where or that*

We accept your suggestion and will edit the manuscript accordingly.

*14) Section 4.2 (pp.12, L16) and Section 5 (pp.13, Ls 28 − 31); This reviewer considers that the automation of PMAF detection might has already been started to be developed based on machine learning technique (e.g., Convolutional Neural Network; CNN). Do you have any opinions on the event search using machine learning? If yes, you also should discuss the relation between your opinion and machine learning technique in the manuscript.*
*There are several recent reports that, with a help of machine learning technique, the auroras detected by the all-sky camera can automatically be categorized. The corresponding links are shown as follows.*
*1.https://www.nature.com/articles/s41598-022-11686-8*

*2.https://tromsoe-ai.cei.uec.ac.jp/#/*
*3.https://www.sciencedirect.com/science/article/pii/S1364682622000797?via%3Dihub#fig1*
   *This reviewer considers that your research results and principle of the PMAF event search can be implemented to these algorithms. However, on the other hand, independently, will the authors build some system to automatically detect the PMAFs in near future?*

Thank you for asking this question. It is true that there has been recent advancement in the effort to automatically detect and categorize aurora. It is the authors impression that these efforts have managed to automatically detect the presence of auroral forms, and in some cases categorization into types of aurora (diffuse, arc, omega etc.). However, as far as we can tell there is currently no automated algorithm that can detect PMAFs. We will mention this in the manuscript.
We might attempt to develop an algorithm to detect PMAFs based on arciness in the near future.

---

## Author Comment (AC2)

**Author Response to RC1**

July 31, 2022

*Major comments:*

*1. One of the major concerns is that the purpose/motivation and new aspects of this paper are unclear. The manuscript does not explain why they examined the morphology PMAFs and why they could examine the morphology "in detail" (e.g., due to new data/method). The term "in detail" is very ambiguous, and the critical point is what those details are. As the authors say in the abstract, previous studies have examined PMAF morphology. The authors should focus more on what is different data/methods and new finding compared with those previous studies.*

Thank you for bringing this up. The motivation behind this paper is that there has not been a study dedicated to the morphological evolution of PMAFs. There have only been previous studies with different focuses that commented on the morphology of PMAFs, however no detailed (as in, in-depth) study of the subject. We have listed the exact advances and novel aspects this paper introduces in response to your first specific comment. We will rework the introduction section to include the information provided in that response.

Furthermore, it is not necessarily the methods and data that are inherently different from previous studies, with the exception of our analysis of the arciness index in relation to PMAF occurrence, but rather the depth of analysis.

*2. The title "Morphology of Poleward Moving Auroral Forms" is too general, and such a title is suitable for a review paper or the first report on morphology. They argued that they found a new feature, merging auroral patches/arcs into larger-scale PMAFs. If it is a really new finding, this paper should focus on it, and the title should be, for example, "Merging of auroral patches/arcs into large-scale PMAFs". More analysis and discussion of the generation mechanism of this phenomenon are needed. Also, I wonder whether this feature is a new typical feature or a particular case of PMAFs?*

Thank you for your comment and suggestion of a new title. We chose this title since this paper is the first paper dedicated to the morphology of PMAFs. And you are right, this topic has been commented on multiple times in previous studies. However, each of those studies were not focused on PMAF morphology.

Moreover, while a major new finding of our paper is the discovery of auroral patches merging before the start of a PMAF event, this is not the only new finding, as there are other important aspects we discuss. Additionally, the arciness analysis is also an important part of our paper which validates our description of PMAF morphology. Thus, it does not seem appropriate to call our paper "Merging of auroral patches into large-scale PMAFs", as that title does not fully describe the breadth of our paper, since it introduces many new points beyond the merging of auroral patches into PMAFs. However, we are surely open to changing the title, provided the new title encompasses all parts of our study. We suggest 'Morphological evolution and spatial profile changes of poleward moving auroral forms'.

We observed the merging of auroral patches into a large-scale arc in less than half of PMAFs in this study. We will make sure this is clearly mentioned in the manuscript.

*3. The analysis of the arcness index is not needed for this paper since it looks like a different topic.*

*Also, it did not help find a new morphology of PMAFs.*

We believe the arciness analysis is a valuable part of this paper. Arciness has been used multiple times in the literature [Partamies et al., 2022, Partamies et al., 2014, Partamies et al., 2017b, Partamies et al., 2017a, Partamies et al., 2015] and has been established as a useful tool to quantify auroral morphology. The major advantage arciness brings to the table is that it is a completely objective measure of auroral morphology. Beyond that, due its quantitative nature, arciness allows us to investigate the morphology of PMAFs in a statistical manner using superposed epoch analyses. Since we are able to explain the evolution of arciness based on our description of PMAF morphology, this validates our report on PMAF morphology.

*Specific comments:*
*P5 section 4.1: The authors abruptly summarize the general morphological evolution without showing any data supporting it. Multiple examples, at least 3-4 cases, showing the general morphological evolution should be displayed before the summary. Also, what is new points that previous works have not been reported?*

We are happy to rearrange section 4.1 to show our summary of the morphological evolution of PMAFs after showing fig. 2 and fig. 3, which combined give three examples of PMAF morphology. Furthermore, fig. 6 gives a fourth example of the morphological evolution of PMAFs. Thus, we believe there are enough specific examples in our manuscript to support our description.
The novel points this paper introduces are numerous small advances and one or two more significant conclusions rather than a single major finding. These new findings are further verified by the use of the arciness analysis, the results of which can be understood based on our description of PMAF morphology.
This paper contributes to the literature by reporting novel aspects on PMAF morphology. One of the novel aspects we report is the observation that PMAFs can form following the merging of distinct auroral patches into a singular arc-like structure, the PMAF. In the literature, the beginning of a PMAF is typically described as an appearance of an auroral arc or a the intensification of the equatorward boundary of the dayside auroral oval. We propose a possible explanation for this phenomenon, which relates to the occurrence of localized dayside magnetopause reconnection, or the localized differences in reconnection efficiencies.
Another new point we report is the intensification of the auroral oval happening at the open-closed boundary, as opposed to the equatorward boundary of the auroral oval. While this difference might seem semantic in nature, because those two domains overlap around magnetic noon, they are not identical. Further away from noon (9-10 and 14-15 MLT), the open closed boundary is inside the dayside auroral oval and PMAFs at those magnetic local times do not emerge from the equatorward boundary of the dayside auroral oval. The equatorward boundary of the auroral oval between dusk/dawn and magnetic noon is on closed field lines, as there can be auroral precipitation from the plasma sheet. The open-closed boundary is poleward of the equatorward boundary and it determines the location at which PMAFs first appear, as they exist exclusively on open field lines.
Furthermore, we also observed PMAFs to frequently expand along the north-south dimension as they propagate poleward, specifically near the end of their lifetime. This has also never been reported before.
We compare some of our conclusions to the conclusion made by Sandholt and Farrugia (2007). They reported a strong dependency in the occurrence of PMAF categories 1 and 2 on IMF By and magnetic local time, where under positive IMF By PMAF2s would almost exclusively occur in the pre-noon sector, while under the same IMF conditions observations of PMAF1s would strongly outnumber those of PMAF2s in the post-noon sector. We have found no such relationship. In our data, we observe approximately 70% PMAF2s and 30% PMAF1s, with no apparent preference for pre-noon or post-noon occurrence under any IMF conditions.
Sandholt and Farrugia also reported observations of poleward boundary intensifications, as opposed to

equatorward boundary intensifications, for PMAFs occurring in the post-noon sector under IMF By>0 and in the pre-noon sector under IMF By<0. While we did not observe intensifications of the auroral oval at its poleward boundary, we did frequently observe intensifications inside the auroral oval and hence poleward of the equatorward boundary. This relates to our previous finding of PMAFs emerging from the open-closed boundary, rather than the equatorward boundary of the auroral oval.

They were also the first to report that PMAF2 events would have a higher green auroral emission component after the rebrigthening event close to the end of the PMAF lifetime. We can confirm this finding based on our data and interpret it according to the proposed re-brightening mechanism proposed by [Fasel et al., 1992].

*Fig3: I cannot see the arc moved poleward. Is this really PMAF event? The author should quantify the velocity of the move of aurora. Also, latitude-longitude grids are needed to identify right direction.*

Thank you for pointing this out. The poleward motion is in fact hard to make out from the ASC image series. This is partly due to the fact that the westward motion is dominant over the poleward propagation. Here is a keogram that includes the event time frame. The black lines show approximate locations of two PMAF events that are visible in the ASC image series in fig. 2. We can include the keogram in our manuscript.

[Figure]

*p9 L2: Where are data supporting "no relationship between PMAF occurrence time and class."?*

Thank you for this question. We have not included any raw data on PMAF class for each event in the manuscript. As an example, here are the PMAFs of event list 2 sorted by PMAF class and start time (times are in UT, MLT ≈ UT + 3):

PMAF1: (06:31, 06:56, 07:11, 07:25, 07:51, 08:12, 08:19, 08:52, 09:25, 09:29)
PMAF2: (06:16, 06:25, 06:50, 07:16, 07:35, 07:46, 07:56, 08:00, 08:39, 09:06, 09:17, 09:35)

Evidently, there is no preference for either class in the pre-/post-noon sector in our data, as has been reported by [Sandholt and Farrugia, 2007]. We will gladly include this data in our manuscript.

*Fig6: Why the author uses several events not all the events?*

Thank you for catching this. These plots were made with all events. However, you are right that the use of the word 'several' is confusing here, and has been changed to 'all'.

*Fig7: Why values of arcness are not shown in the figure? P12 L14: Where are data supporting "the general evolution of arciness relating to PMAF occurrence."?*

That paragraph refers to the general evolution of arciness over the course of PMAF events, and how that evolution relates to the general evolution of the morphology of PMAFs. We use fig. 7 as an example to map actual ASC images to approximate times in the arciness evolution (fig. 6) and to give another example of PMAF morphology. Hence, the data is contained in fig. 6. We do not show arciness values for the ASC images in fig. 7 as it is merely an example, and cannot be used to validate the results of a statistical analysis, the results of which are plotted in fig. 6.

**References**

[Fasel et al., 1992] Fasel, G. J., Minow, J. I., Smith, R. W., Deehr, C. S., and Lee, L. C. (1992). Multiple brightenings of transient dayside auroral forms during oval expansions. *Geophysical Research Letters*, 19(24):2429–2432.

[Partamies et al., 2015] Partamies, N., Juusola, L., Whiter, D., and Kauristie, K. (2015). Substorm evolution of auroral structures. *Journal of Geophysical Research: Space Physics*, 120(7):5958–5972.

[Partamies et al., 2017a] Partamies, N., Weygand, J. M., and Juusola, L. (2017a). Statistical study of auroral omega bands. *Annales Geophysicae*, 35(5):1069–1083.

[Partamies et al., 2017b] Partamies, N., Whiter, D., Kadokura, A., Kauristie, K., Tyssøy, H. N., Massetti, S., Stauning, P., and Raita, T. (2017b). Occurrence and average behavior of pulsating aurora. *Journal of Geophysical Research: Space Physics*, 122(5):5606–5618.

[Partamies et al., 2022] Partamies, N., Whiter, D., Kauristie, K., and Massetti, S. (2022). Local time dependence of auroral peak emission height and morphology. *Ann. Geophys. Discuss.*

[Partamies et al., 2014] Partamies, N., Whiter, D., Syrjaesuo, M., and Kauristie, K. (2014). Solar cycle and diurnal dependence of auroral structures. *Journal of Geophysical Research: Space Physics*, 119(10):8448–8461.

[Sandholt and Farrugia, 2007] Sandholt, P. E. and Farrugia, C. J. (2007). Poleward moving auroral forms (PMAFs) revisited: responses of aurorae, plasma convection and birkeland currents in the pre- and postnoon sectors under positive and negative IMF by conditions. *Annales Geophysicae*, 25(7):1629–1652.

---

## Author Response (AR1)

**Changes to the PMAF Morphology manuscript**

September 28, 2022

**Title:**

Our suggested new title is: "Morphological evolution and spatial profile changes of poleward moving auroral forms"

**Abstract:**

Reworded the last part of the abstract to more clearly state the motivation of this study

**Introduction:**

Stated that we do not investigate the driver of PMAFs and assume they are caused by reconnection Reworked motivation paragraph to improve clarity Moved station coordinates to instrumentation section

**Methodology:**

Revised explanation of the compiling of the event lists clearly Reformulated explanation of SEA

**Results and Discussion:**

Rearranged section so that summary is at the end after going through examples in fig. 2 and fig. 3 Replaced solar wind temperature values with solar wind number density (table 1) Included raw data on PMAF class and occurrence time (table 2) Revised explanation of our reasoning behind re-branding the term EBI to OCBI Included our reasoning behind a possible explanations for the appearance of green auroral patches preceding the formation of a PMAF Added timestamps and figure header to fig. 2 and 3 Amended fig. 4 legend to explain colors Changed text referring to fig. 5: histogram is black not red. Amended fig. 6 legend to explain labels in figure Reworked the fig. 7 explanation to include fig. 6 Expanded on automated auroral detection techniques/shortcoming in regards to PMAF detection

**Conclusion**

Replaced "morphological evolution" with "temporal and spatial profile changes" pp.13, L5 and L11 Replaced "which" with "that" pp.13, L12

**Supplementary Information**

Added .gif file showing fig. 2 and fig. 3 animated

**Comments**

Regarding the comment about investigating the drivers of our events: We have checked SuperDARN and EISCAT data for our events and have not found any usable data. Furthermore, we also focus on the spatial profile evolution of PMAFs, rather than their driving forces.

---

## Referee Report (RR1)

2[nd] round reviewer report of "Morphological Evolution and Spatial Profile Changes of Poleward Moving Auroral Form (previously entitled by "The Morphology of Poleward Moving Auroral Forms") by Anton Goertz, Noora Partamies, Daniel Whiter, and Lisa Baddeley.

 This reviewer thanks all authors for carefully replying to my questions and making major revisions along my suggestions and corrections. This reviewer seems that this paper gets close to the publication to Annales Geophysicae but still have the following five minor questions, suggestions, and comments on the revised manuscript.

I. Page 3, the paragraph starting with "For each PMAF event…": the authors should re-organize this paragraph so that the two reasons to perform the analyses separately based on two event lists in this study can "parallel" and "clearly" be described.
 The first reason should be "Although the 1[st] database contains 23 PMAFs that occurred between 2003 and 2008, PMAF took place on 10 different days, hence, some of those days had only 1 PMAF event, while on other days multiple PMAFs occurred. The second event list contains 18 PMAFs that all occurred on the same. This is unusual and thus we separate the two databases and analyze them separately." The second reason is "the camera setups, including different cameras and present/absent narrow-band filter, used in this study, are different in the two databases". Above information should be included in this paragraph and re-organized.

II. On the definition of "re-brightening events"; this referee can understand what "re-brightening events" are with the authors' explanations. However, I cannot understand where this detail was described. In your reply, "This is mentioned in the manuscript on page 8 line6…" but, I cannot find these sentences there. Please re-check this.

III. Page 6, L.8: At least, this reviewer has never heard the terminology of "equatorward boundary intensification (EBI)" (but, instead, poleward boundary intensification (PBI) is frequently used and widely known). Although the authors say, "This phenomenon is commonly referred to as an EBI in the literature…", which literature(s) is (are) EBI shown and discussed? Please show the citations.

IV. Page 7: "…the authors believe…" → "…we believe that it is …"
"the authors" should not be used in the manuscript but should be replaced with "we". Please check whether or not you are making this usage elsewhere in the manuscript.

V. Location of quotation: "While there have been recent advancements in the automation of detection… of auroral forms (Nanjo et al., 2022; Guo et al., 2022), there is currently no automated…"

---

## Author Response (AR2)

**Author Response**

**December 20, 2022**

**1  RC1**

*Specific comments on the revised manuscript:*
*P1 L15: The phase of "While there have been previous studies commenting on PMAF morphology" can be removed.*

Thank you for your comments, and you are right, this has been removed.

*P2 L23: "commented on their morphological evolution." can be removed. An important issue is not whether they commented on the morphology or not, but what they show the morphology.*

You are right, this has been removed.

*P3 L1: What does MSP stand for?*

The acronym MSP stands for Meridian scanning photometer, which we have added to the manuscript.

*P3 L41: be commented on in the discussion. ---> be discussed later.*
*P3 P41:"It is also one of two reasons we analyze the two event lists separately in the quantitative part of this study." I do not understand what this sentence means. What is the first reason?*

The two reasons are: 1) The unusually high PMAF occurrence rate 2) Different camera setups during the two event lists. We have adjusted the manuscript to make this clear.

*P4 L30:"The first complete description of the morphological evolution. . ." can be removed. If it was the complete description, no need to be investigated in detail anymore.*

We think you mean P2 L30 instead of P4 L30. And we agree with your assessment, the phrase has been removed.

**2  RC2**

*I. Page 3, the paragraph starting with "For each PMAF event. . .": the authors should re-organize this paragraph so that the two reasons to perform the analyses separately based on two event lists in this study can "parallel" and "clearly" be described. The first reason should be "Although the 1 st database contains 23 PMAFs that occurred between 2003 and 2008, PMAF took place on 10 different days, hence, some of those days had only 1 PMAF event, while on other days multiple PMAFs occurred. The second event list contains 18 PMAFs that all occurred on the same. This is unusual and thus we separate the two databases and analyze them separately." The second reason is "the camera setups,*

*including different cameras and present/absent narrow-band filter, used in this study, are different in the two databases". Above information should be included in this paragraph and re-organized.*

Thank your for your comment. This paragraph has been reworked following this and the other reviewer's comment.

*II. On the definition of "re-brightening events"; this referee can understand what "re-brightening events" are with the authors' explanations. However, I cannot understand where this detail was described. In your reply, "This is mentioned in the manuscript on page 8 line6..." but, I cannot find these sentences there. Please re-check this.*

We apologize for the confusion, the actual location of this sentence is page 6 line 2-3 (revised manuscript).

*III. Page 6, L.8: At least, this reviewer has never heard the terminology of "equatorward boundary intensification (EBI)" (but, instead, poleward boundary intensification (PBI) is frequently used and widely known). Although the authors say, "This phenomenon is commonly referred to as an EBI in the literature...", which literature(s) is (are) EBI shown and discussed? Please show the citations.*

EBIs have been commonly reported to be observed during the onset of a PMAF event. We have added multiple citations to this claim.

*IV. Page 7: "...the authors believe..." "...we believe that it is ..." "the authors" should not be used in the manuscript but should be replaced with "we". Please check whether or not you are making this usage elsewhere in the manuscript.*

We accept your suggestion and have changed our use of the phrase "authors" with "we" in all instances.

*V. Location of quotation: "While there have been recent advancements in the automation of detection... of auroral forms (Nanjo et al., 2022; Guo et al., 2022), there is currently no automated..."*

We accept your suggestion and have moved the location of the citation.